# Xpert Ultra stool testing to diagnose tuberculosis in children in Ethiopia and Indonesia: a model-based cost-effectiveness analysis

Nyashadzaishe Mafirakureva [ID],[1] Eveline Klinkenberg,[2,3] Ineke Spruijt,[4] Jens Levy,[4] Debebe Shaweno,[1] Petra de Haas,[4] Nastiti Kaswandani,[5] Ahmed Bedru,[6] Rina Triasih,[7] Melaku Gebremichael,[6] Peter J Dodd,[1] Edine W Tiemersma [ID] [4]

For numbered affiliations see end of article.

**Correspondence to**
Dr Nyashadzaishe Mafirakureva;
n.mafirakureva@sheffield.ac.uk

## ABSTRACT

**Objectives** The WHO currently recommends stool testing using GeneXpert MTB/Rif (Xpert) for the diagnosis of paediatric tuberculosis (TB). The simple one-step (SOS) stool method enables processing for Xpert testing at the primary healthcare (PHC) level. We modelled the impact and cost-effectiveness of implementing the SOS stool method at PHC for the diagnosis of paediatric TB in Ethiopia and Indonesia, compared with the standard of care.

**Setting** All children (age <15 years) presenting with presumptive TB at primary healthcare or hospital level in Ethiopia and Indonesia.

**Primary outcome** Cost-effectiveness estimated as incremental costs compared with incremental disability-adjusted life-years (DALYs) saved.

**Methods** Decision tree modelling was used to represent pathways of patient care and referral. We based model parameters on ongoing studies and surveillance, systematic literature review, and expert opinion. We estimated costs using data available publicly and obtained through in-country expert consultations. Health outcomes were based on modelled mortality and discounted life-years lost.

**Results** The intervention increased the sensitivity of TB diagnosis by 19–25% in both countries leading to a 14–20% relative reduction in mortality. Under the intervention, fewer children seeking care at PHC were referred (or self-referred) to higher levels of care; the number of children initiating anti-TB treatment (ATT) increased by 18–25%; and more children (85%) initiated ATT at PHC level. Costs increased under the intervention compared with a base case using smear microscopy in the standard of care resulting in incremental cost-effectiveness ratios of US$132 and US$94 per DALY averted in Ethiopia and Indonesia, respectively. At a cost-effectiveness threshold of 0.5×gross domestic product per capita, the projected probability of the intervention being cost-effective in Ethiopia and Indonesia was 87% and 96%, respectively. The intervention remained cost-effective under sensitivity analyses.

**Conclusions** The addition of the SOS stool method to national algorithms for diagnosing TB in children is likely to be cost-effective in both Ethiopia and Indonesia.

### STRENGTHS AND LIMITATIONS OF THIS STUDY

⇒ The first study to evaluate the impact and cost-effectiveness of including the simple one-step method to the existing national algorithms for diagnosing tuberculosis (TB) in children.
⇒ The evaluation used a systematic literature review to inform model parameters.
⇒ The analysis is based on two very diverse settings and is likely to have global relevance to countries with high TB burden.
⇒ Limited availability of local data to inform some important parameters, including referral rates and primary cost data.
⇒ Patient costs were not included, meaning any patient benefits from reduced referrals were not captured.

## BACKGROUND

It was estimated that in 2018, around 1.1 million children below 15 years of age fell ill from tuberculosis (TB).[1] In the same year, 250 000 children died of TB, mostly because TB was not diagnosed or was diagnosed too late. It is estimated that 55% of TB cases are missed, particularly in the youngest age group.[2] TB in children presents with nonspecific signs and symptoms, and *Mycobacterium tuberculosis* bacilli are usually not detected.[2] Partly, this is because the main specimen used for diagnosing pulmonary TB is sputum, which is challenging to obtain, especially from young children. Therefore, (semi-)invasive methods such as nasogastric aspiration and sputum induction are often required. These methods are painful and stressful for children and caregivers and sometimes require hospitalisation. Moreover, not all primary healthcare (PHC) facilities in TB endemic areas, where parents with children usually first seek care, have facilities and qualified saff to perform these procedures. Alternative, non-invasive specimens, such as stool, can

be used for the diagnosis of TB in children using Xpert MTB/RIF (Xpert) technology.[3 4]

Since January 2020, WHO recommends Xpert testing of stool specimens as a primary diagnostic test for TB in children with signs and symptoms of pulmonary TB.[5] This recommendation has the potential to improve bacteriological confirmation of TB in children, and is increasingly being adopted by national TB programmes, for example, Ethiopia.[6] However, to make the test fit for use at the PHC level, a simple, non-hazardous and cheap method to process stool for Xpert testing was needed. Several centrifuge-free methods have been proposed,[7–10] but all need additional equipment and/or consumables which may not be (easily) available in peripheral lower-level public health facilities. Therefore, we developed a simple one-step (SOS) stool processing method for Xpert testing. This method can be applied in any laboratory with an Xpert machine, as it does not require additional equipment or consumables than those delivered routinely with the Xpert cartridges.[11] Limited preliminary data suggest that Xpert Ultra on stool samples processed using the SOS stool method has higher sensitivity compared with other stool processing methods. Available systematic reviews on the diagnostic accuracy of stool testing have reported sensitivity of 50–67%.[3 4 12] The variation in sensitivity estimates may be explained by a variation in studies included, and thus, variation in study populations, stool processing methods and reference standards (sputum culture[4 12] or a combination of sputum culture and sputum Xpert[3]) included in each review. This method has the potential to significantly impact the number of children receiving a bacteriological confirmation of TB, including rifampicin resistance profile. Consequently, more paediatric TB patients can be diagnosed at lower levels of the healthcare system, with a reduced time to diagnosis because no referrals to higher levels of healthcare are needed as well as reduced costs for both the healthcare system and families.

However, evidence on the impact and cost-effectiveness of the SOS Xpert stool processing method is needed to inform implementation and scale-up in routine healthcare systems. Therefore, we modelled the potential impact and cost-effectiveness of bringing this test to the lower healthcare level where children present first, focusing on Ethiopia and Indonesia. Specifically, we aimed to estimate the impact of implementing the Xpert stool test for the diagnosis of pulmonary TB among children at the PHC level on rates of bacteriological confirmation of TB and mortality among children, the costs to the healthcare provider, and the incremental cost-effectiveness of the approach. Ethiopia and Indonesia are currently among the 30 high TB burden countries in the world.[13] The incidence of TB was estimated to be 301 (276–328) per 100 000 population with 824 000 (755 000–897 000) people falling ill with TB in 2020 in Indonesia. In Ethiopia, the incidence of TB was estimated to be 132 (92–178) per 100 000 population with 151 000 (106 000–205 000) people falling ill with TB in 2020. While TB

diagnosis and treatment in Ethiopia largely occur in the public sector, the private sector plays a substantial role in Indonesia.

## METHODS
### Conceptual approach
We developed a conceptual model of care pathways for children (age <15 years) with presumptive TB presenting at either PHC facilities or hospitals, referral (including self-referral) between these levels, and clinical and bacteriological assessment and reassessment (see figure 1). This description of stages in patient care was based on national TB guidelines and local knowledge, and were broad enough to capture pathways in both Ethiopia and Indonesia, and incorporate the standard of care (SOC) as well as the intervention.

We defined patients with presumptive TB, following clinical guidelines in both settings, as children with signs or symptoms suggestive of pulmonary TB (at least one) such as persistent cough, unexplained fever and/or night sweats, poor weight gain or weight loss, reduced playfulness or malaise, history of contact with a TB patient, or enlarged lymph nodes in the neck (Ethiopia only).

Under the SOC, national guidelines in both countries recommend the use of GeneXpert for the diagnosis of paediatric TB.[6 14] However, in both countries, sputum smear microscopy (SSM) is allowed for diagnosis if the PHC has no access to GeneXpert. Despite this recommendation, in both countries, most PHC units do not have access to a GeneXpert machine, and therefore use SSM for the diagnosis of paediatric TB. For the diagnosis of paediatric TB, in the primary (base-case) analyses, we assumed that SSM was the bacteriological test used at PHC in SOC in both Ethiopia and Indonesia. In the sensitivity analyses, we considered alternate scenarios with Xpert used for bacteriological testing for sputum in SOC.

The intervention was modelled as implementing the simple stool Xpert testing method at PHC and hospital level. Thus, considering spontaneous sputum expectoration to be limiting in obtaining a test result under SOC, we conceptualised the intervention as increasing the fraction of children with a bacteriological test result at both the primary and higher healthcare level.

We assumed that children with a negative bacteriological test under the intervention would receive clinical assessments for TB while only a small proportion would get clinical assessments under SOC. A clinical diagnosis can be made based on TB-suggestive signs or symptoms, chest X-ray results and tuberculin skin test (Indonesia only) or contact history with a TB patient. Indonesian referral centres that do not have access to chest X-rays and/or TB skin tests use a score chart for clinical diagnosis.

Systematic review data on the sensitivity of stool-based diagnostics for identifying TB in children, indicate sensitivity of 50–67%[3 4 12] in children with bacteriologically confirmed TB, but very poor sensitivity (2–6%) in clinically diagnosed TB.[3 4] We, therefore, assumed that stool

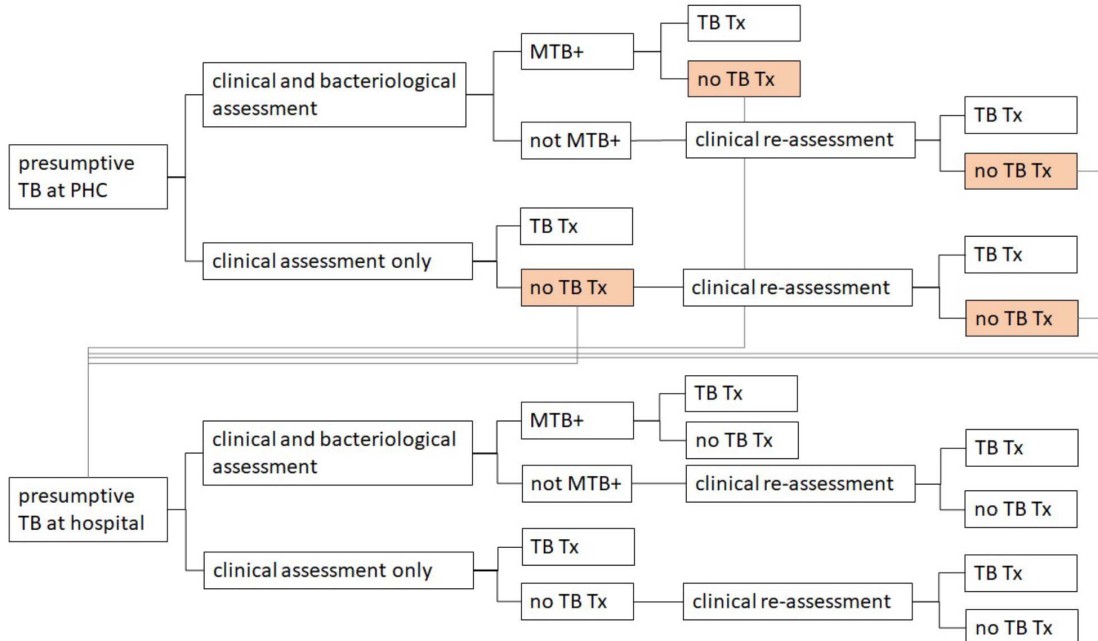

**Figure 1** Simplified diagram of decision-analytical model showing the pathways of care for TB diagnosis and treatment. The decision tree shows children with presumptive TB presenting at either PHC facilities or hospitals where they undergo clinical evaluation with or without bacteriological testing. All children diagnosed with TB are considered for anti-TB treatment. Children with a negative bacteriological test or those not initially diagnosed with TB after clinical assessment only can be reassessed clinically. Coloured boxes depict the potential of referral to a higher-level facility and referrals (indicated by grey lines) from PHC to hospital for further assessment can occur for children without a diagnosis of TB. Each pathway extends to death or survival, however, these details are omitted here to keep the diagram simple. See online supplemental appendix 2A for more details on the pathway and parametrisation of the model. MTB, *Mycobacterium tuberculosis*; PHC, primary healthcare; TB, tuberculosis; TB Tx, TB diagnosis and anti-TB treatment.

testing would only detect a proportion of those children who would be bacteriologically positive under ideal circumstances. The accuracy of Xpert testing on stool using the SOS method was modelled based on a systematic review[4] which reported pooled sensitivity and specificity of stool Xpert of 57.1% (95% CI 51.5-62.7%) and 98.1% (95% CI 97.5-98.6%), respectively, compared with culture on a respiratory sample as the reference standard. The intervention was to reduce mortality through higher sensitivity for detecting TB, and to reduce referrals and reassessments.

### Modelling approach
The pathway of care shown in figure 1 was coded into a decision tree using the HEdtree package in R.[15 16] Referral endpoints from PHC level were modelled by adding an identical hospital care pathway to follow the three paths for referral from PHC level. All care outcomes were extended to either death or survival. The probability of children following different pathways through the tree was assumed to depend on: true TB status and age (0–4 years or 5–14 years). Mortality risk from TB by age group and anti-TB treatment (ATT) status was modelled using a published approach,[17] using case-fatality ratios based on systematic review data.[18] We neglected mortality in children who were truly negative for TB. We did not model drug-resistant TB or HIV status.

All parameters in the model were treated as uncertain and following specified distributions. All results were based on applying the model to calculate mean outcomes from the tree for each of 10000 samples from these parameter distributions.

### Literature review and parameterisation
To inform the parameters needed in the decision tree model (figure 1, parameters noted as such in table 1), we followed a three-step data collection process. First, we reviewed data from ongoing studies in Ethiopia[11] and Indonesia (Kaswandani *et al* Xpert MTB/RIF testing on stools using simple preprocessing methods to diagnose childhood pulmonary tuberculosis in Indonesia. 2019). Second, we systematically searched peer-reviewed literature for parameters not available from country study data. In brief, initially, systematic reviews on TB in children were sought by a search of PubMed including the terms 'systematic review', 'meta-analysis', 'tuberculosis' and 'children' on 19 June 2020. Subsequently, we constructed pooled estimates from primary literature, published from 2010 to present, about TB diagnostic testing in infants and children, including healthcare seeking and healthcare cascade with a focus on Ethiopia and Indonesia. For this, a systematic search strategy was developed by an information specialist combining free-text and thesaurus searching. Except for searches specifically addressing

**Table 1** Table of parameters used in modelling and underlying evidence

| Description | Source | References | Mean (IQR) |
|---|---|---|---|
| Sensitivity of Xpert on stool in bacteriologically positive children | Existing review | Mesman et al 2019[4] | 0.571 (0.515–0.627) |
| Specificity of Xpert on stool in bacteriologically positive children | Existing review | Mesman et al 2019[4] | 0.981 (0.975–0.986) |
| Sensitivity of Xpert on sputum in C+ | Existing review | Detjen et al 2015[35] | 0.621 (0.582–0.659) |
| Specificity of Xpert on sputum in C+ | Existing review | Detjen et al 2015[35] | 0.980 (0.977–0.984) |
| Sensitivity of SM on sputum in C+ | Existing review | Detjen et al 2015[35] | 0.257 (0.215–0.302) |
| Specificity of SM on sputum in C+ | Existing review | Detjen et al 2015[35] | 0.995 (0.994–0.997) |
| Spontaneous sputum possible (0–4 years) | Our review | see online supplemental appendix 2A | 0.024 (0.020–0.027) |
| Spontaneous sputum possible (5–14 years) | Our review | see online supplemental appendix 2A | 0.377 (0.254–0.512) |
| Fraction of children bacteriologically confirmable <5 years | Our review | see online supplemental appendix 2A | 0.380 (0.363–0.397) |
| Fraction of children bacteriologically confirmable 5–14 years | Our review | see online supplemental appendix 2A | 0.684 (0.659–0.711) |
| Prevalence of true TB in presumptive | Our review | see online supplemental appendix 2A | 0.453 (0.289–0.607) |
| Specificity of clinical diagnosis <5 years | Our review | Marais 2006 (see online supplemental appendix 2A) | 0.928 (0.908–0.945) |
| Sensitivity of clinical diagnosis <5 years | Our review | Marais et al 2006[36] | 0.518 (0.482–0.554) |
| Specificity of clinical diagnosis 5–14 years | Our review | Marais et al 2006[36] | 0.901 (0.878–0.921) |
| Sensitivity of clinical diagnosis 5–14 years | Our review | Marais et al 2006[36] | 0.627 (0.592–0.661) |
| Proportion of first care-seeking at PHC for Ethiopia | Our review | Fekadu et al 2017[37] | 0.896 (0.777–0.973) |
| Proportion of first care-seeking at PHC for Indonesia | Our review | Surya et al 2017[38] | 0.928 (0.801–0.992) |
| Fraction of presumptive TB under 5 years Ethiopia | Routine data | fraction of WHO TB <5 | 0.371 (0.300–0.447) |
| Fraction of presumptive TB under 5 years Indonesia | Routine data | fraction of WHO TB <5 | 0.514 (0.485–0.543) |
| Referral PHC ->Hospital after clinical re-assessment following bacteriological negative result Ethiopia | Expert opinion | see online supplemental appendix 2A | 0.045 (0.019–0.088) |
| Referral PHC ->Hospital after clinical re-assessment following bacteriological negative result Indonesia | Expert opinion | see online supplemental appendix 2A | 0.200 (0.107–0.272) |
| Referral PHC ->Hospital after initial clinical assessment without bacteriological test result Ethiopia | Expert opinion | see online supplemental appendix 2A | 0.800 (0.728–0.899) |
| Referral PHC ->Hospital after initial clinical assessment without bacteriological test result Indonesia | Expert opinion | see online supplemental appendix 2A | 0.500 (0.391–0.607) |
| Clinical reassessment, PHC Ethiopia | Expert opinion | see online supplemental appendix 2A | 0.045 (0.019–0.088) |
| Clinical reassessment, PHC Indonesia | Expert opinion | see online supplemental appendix 2A | 0.045 (0.019–0.088) |
| Proportion of bacteriologically confirmed children initiating anti-TB treatment, PHC | Assumption | | 0.953 (0.937–0.966) |
| Proportion of bacteriologically confirmed children initiating anti-TB treatment, hospital | Assumption | | 0.953 (0.937–0.966) |
| Clinical reassessment after bacteriologically negative, PHC | Assumption | | 0.045 (0.019–0.088) |
| Clinical reassessment after bacteriologically negative, hospital | Assumption | | 0.045 (0.019–0.088) |
| Clinical reassessment, hospital | Assumption | | 0.045 (0.019–0.088) |
| Referral PHC ->hospital after clinical re-assessment without bacteriological test result | Assumption | | 0.500 (0.391–0.607) |
| CFR children <5 years on TB treatment | Existing review | Jenkins et al 2017[18] | 0.019 (0.012–0.029) |
| CFR children 5–14 years on TB treatment | Existing review | Jenkins et al 2017[18] | 0.008 (0.006–0.011) |
| CFR children <5 years without TB treatment | Existing review | Jenkins et al 2017[18] | 0.436 (0.413–0.460) |
| CFR children 5–14 years without TB treatment | Existing review | Jenkins et al 2017[18] | 0.149 (0.137–0.162) |

More details on parameter distributions, parameter naming and methods are available in online supplemental appendix 2A.
C+, culture positive; CFR, case fatality rate; PHC, primary health care; SM, smear microscopy; TB, tuberculosis.

Indonesia/Ethiopia, we excluded case reports, non-English and non-human studies, and papers with terms for Bacillus Calmette-Guerin (BCG), latent TB, gamma interferon (IFN-γ) release assay (IGRA) and tuberculin skin test in titles because of their relevance to TB infection, not active pulmonary TB. Searches were conducted between 19 October 2020 and 26 October 2020. Finally, to inform remaining parameters, we sought opinion from TB experts from each country (authors AB and MG for Ethiopia and NK and RT for Indonesia) in an iterative process using a questionnaire, and remote workshops to explain the model and focus on parameters identified as influential by one-way sensitivity analysis. More details are provided in online supplemental appendix 1 (literature search) and online supplemental appendix 2A (model parameter estimation).

### Cost parameters and health economic approach

We collected costs (reported in 2019 USD) from the healthcare provider's perspective and adjusted historical costs for inflation to 2019 prices using relevant gross domestic product (GDP) deflators.[19] We transferred costs from other countries to Ethiopia and Indonesia by applying relevant purchasing power parity conversion factors.[20] All costs were assumed to accrue in the present, with no discounting applied.

We assumed the cost for the initial TB assessment at the PHC was equivalent to the country-specific cost of two outpatient visits (or a single outpatient visit for reassessment) to a health centre (health centre with no beds from WHO-CHOICE estimates[21]). Similar assumptions were used for hospital assessment and reassessment with the corresponding WHO-CHOICE cost estimates.[21] The cost of bacteriological investigation in the SOC includes the country specific unit cost of either SSM[22 23] or Xpert, depending on availability at each level of care, adding the unit costs for collecting two sputum samples for testing with SSM or one sample for testing with Xpert. The unit costs for Xpert were estimated based on country specific data available from the OneHealth Tool (see online supplemental appendix 2B).[24] Country-specific unit costs for collecting sputum samples were not available and are based on a study done in adults from South Africa.[25] In the intervention, we applied the unit cost for collecting a single stool sample based on estimates provided by the Paediatric Operational Sustainability Expertise Exchange group.[26] Treatment cost for diagnosed TB comprises the cost of anti-TB drugs (including pyridoxine), from the Global Drug Facility,[27] the costs of follow-up visits (drug pickups or medical review) according to national TB treatment guidelines at the healthcare facilities based on WHO-CHOICE unit cost estimates, and the costs of laboratory monitoring in bacteriologically confirmed TB only (see online supplemental appendix 2B (overview of cost parameters)).

We used a disability-adjusted life-year (DALY) framework, calculating the life-years saved over a life-time horizon with a discount rate of 3% based on United Nations Population Division country-specific life tables. A simple mean across ages included in the 0–4 and 5–14 year age groups was used, and decrements in health-related quality of life or subsequent survival were not modelled.

### Metrics calculated

For every 100 children seeking care with presumptive TB in each country, we calculated the deaths, DALYs, costs, referrals, clinical assessments, bacteriological assessments, ATTs, percent of true TB receiving ATT, percent of those receiving ATT bacteriologically confirmed, percent of those receiving ATT initiated at PHC, percent of ATT that is false-positive, as well as the change in these quantities under the intervention. We report the incremental cost-effectiveness ratio (ICER). For each country, we produced plots of the cost-effectiveness plane, cost-effectiveness acceptability curve, and expected net benefit, and tornado plots illustrating the one-way sensitivity of outcomes to influential model parameters. We also undertook specific scenario analyses: (1) we considered a low TB prevalence scenario (half the base-case prevalence among presumptive TB patients); (2) we considered Xpert as the universally available bacteriological test instead of sputum-smear microscopy in SOC; (3) we considered discount rates of 0% and 5% for the life-years. The results of these sensitivity analyses are included in online supplemental appendix 3 (additional results). Results are presented following the Consolidated Health Economic Evaluation Reporting Standards Statement (online supplemental appendix 4).

### Patient and public involvement

Study participants or the public were not involved in the design, or conduct, or reporting, or dissemination plans of our research.

### RESULTS

Following our review of the literature, we developed the model parametrisation shown in table 1. The data sources and approach to synthesis for each parameter are described in detail, respectively, in online supplemental appendices 1 and 2A. Country-specific data were used to inform the proportion of children submitting a spontaneously expectorated sputum sample, the fraction of presumptive TB in children under 5 years, and the level of initial care-seeking at PHC. We used existing systematic reviews for the basis of parameters describing diagnostic test accuracy, our own pooled estimates of true TB prevalence among presumptive TB, the fraction of TB that is bacteriological confirmable and the fraction of children able to spontaneously expectorate. Evidence for the accuracy of clinical diagnosis was limited, and published evidence was completely lacking for other parameters around referral and reassessment. Hence, we based these on expert opinion. Unit costs used in the analysis are shown in table 2.

**Table 2** Unit costs for different activities

| Cost description | Unit cost, US$ (SD) | |
| --- | --- | --- |
| | Ethiopia | Indonesia |
| TB assessment at health centre | 10.22 (5.29) | 43.35 (24.24) |
| TB reassessment at health centre | 5.11 (2.25) | 21.68 (10.52) |
| Self-expectorated sputum sample | 2.32 (0.58) | 1.74 (0.43) |
| Stool sample | 1.67 (0.42) | 1.67 (0.42) |
| Sputum smear microscopy examination | 3.39 (1.44) | 7.54 (1.58) |
| GeneXpert test | 26.04 (7.09) | 23.70 (7.11) |
| TB treatment at health centre | 398.74 (177.22) | 161.03 (78.59) |
| TB assessment at hospital | 14.37 (6.59) | 61.00 (30.23) |
| TB reassessment at health centre | 5.11 (2.25) | 21.68 (10.52) |
| TB treatment at hospital | 548.46 (208.38) | 213.98 (91.47) |

See online supplemental appendix 2B for methods and naming conventions.
SD, Standard deviation; TB, tuberculosis.

The intervention increased the sensitivity to detect true TB by over 10 percentage points in each country and resulted in around a fourfold increase in the proportion of patients with TB diagnosed that are bacteriologically confirmed. Specificity showed little change under the intervention (<1% change). In both countries, the proportion of children referred (or self-referred) to higher levels of care after seeking care at PHC level fell by more than twofold. In both countries, the average total number of assessments for children with presumptive TB increased from around 2 per child under SOC to around 2.5 per child with the intervention, and the total number of bacteriological investigations increased more than threefold (table 3).

The relative number of children initiated on ATT increased by 19–25% under the intervention. A larger fraction (~40% relative increase) of children received ATT with the intervention, and more children (~10% point increase) initiated ATT at PHC level (table 3). Restricting to children under 5, we found bigger increases in the number of bacteriological investigations (+30-fold), and the proportion of TB cases diagnosed that are bacteriologically confirmed (+50%). We also found a larger reduction in referrals of children with presumptive TB to higher levels of care in both countries (almost threefold) (see online supplemental appendix 3 (additional results)).

In both countries, the increase in sensitivity of a TB diagnosis under the intervention generated a corresponding reduction in mortality: a 14–20% relative reduction in the fraction of children with presumptive TB dying (table 3). In both countries, costs increased under the intervention, and the base-case (using smear microscopy in the SOC) ICERs were US$132/DALY averted in Ethiopia and US$94/DALY averted in Indonesia (figure 2). Restricting the analysis to children under 5 years resulted in cost savings with ICERs of US$-78/DALY averted in Ethiopia

and increased the ICER to US$209/DALY averted in Indonesia.

## Uncertainty and sensitivity analyses

Model projections showed large uncertainty (figure 2) that included cost savings under intervention (25% of the runs for Ethiopia and 28% for Indonesia), but also some increases in mortality (1.2% of the runs for Ethiopia and 2.8% for Indonesia). At a cost-effectiveness threshold of 0.5×GDP our analysis projected a probability of being cost effective of 87% in Ethiopia and a 96% in Indonesia (see online supplemental appendix 3). The corresponding probabilities for a 1×GDP threshold were 95% (Ethiopia) and 97% (Indonesia). Tornado plots (figure 3) show that prevalence of true TB among presumptive TB, the sensitivity of stool, and the fraction of children able to expectorate were the largest drivers of uncertainty (see also online supplemental appendix 3).

Under the assumption that Xpert was used in the SOC, the ICERs were US$138/DALY averted in Ethiopia and US$115/DALY averted in Indonesia. Assuming half the prevalence of true TB among patients with presumptive TB changed the ICERs to US$145/DALY averted in Ethiopia and US$150/DALY averted in Indonesia. Finally, assuming a 0% discount rate changed the ICERs to US$55/DALY averted in Ethiopia and US$38/DALY averted in Indonesia, whereas 5% discount rate generated ICERs of US$199/DALY averted in Ethiopia and US$142/DALY averted in Indonesia.

## DISCUSSION

In this modelling analysis, we found that the introduction of routine Xpert stool-based diagnostics (using the SOS method) was cost-effective in both Ethiopia and Indonesia. In the context of predominantly clinical diagnosis of TB in children, particularly among those

**Table 3** Outcomes per 100 children seeking care under standard of care (SOC) and intervention (INT) in each country

| Quantity per 100 children with presumptive TB (unless stated) | Ethiopia | | | Indonesia | | |
|---|---|---|---|---|---|---|
| | SOC | INT | Difference | SOC | INT | Difference |
| Children with true TB | 45.5 (8.7–85.0) | 45.5 (8.7–85.0) | 0.0 (0.0–0.0) | 45.5 (8.7–85.0) | 45.5 (8.7–85.0) | 0.0 (0.0–0.0) |
| Assessments | 201.8 (171.8–230.9) | 246.2 (207.3–283.5) | 44.4 (29.5–58.1) | 204.2 (173.4–233.5) | 249.9 (211.2–286.5) | 45.7 (31.9–58.0) |
| Bacteriological investigations | 30.7 (8.7–57.5) | 102.3 (86.8–112.0) | 71.7 (41.5–96.3) | 24.7 (7.8–43.2) | 103.0 (87.5–112.6) | 78.2 (54.5–98.4) |
| Anti-TB treatments (ATT) | 32.2 (13.2–54.5) | 40.3 (17.6–64.4) | 8.1 (0.6–20.3) | 33.3 (14.1–55.3) | 39.5 (17.1–63.3) | 6.2 (0.1–15.2) |
| ATT initiated at PHC† | 71.8 (62.3–79.6) | 81.9 (71.6–89.5) | 10.1 (5.8–14.2) | 73.0 (63.2–80.3) | 84.4 (73.2–91.2) | 11.3 (7.1–15.4) |
| Percent of true TB receiving ATT† | 58.3 (43.0–71.1) | 73.0 (66.7–78.8) | 14.7 (2.8–30.5) | 60.3 (48.2–71.4) | 71.8 (65.9–77.3) | 11.5 (1.8–23.1) |
| Percent of ATT bacteriologically confirmed† | 8.0 (1.7–19.8) | 32.8 (20.7–44.1) | 24.8 (10.6–37.8) | 5.9 (1.4–12.9) | 32.5 (20.9–43.4) | 26.6 (14.9–38.2) |
| Percent of ATT false-positive† | 21.9 (2.8–64.6) | 21.9 (2.9–64.9) | 0.0 (–3.0 to 4.0) | 22.0 (2.8–65.1) | 21.8 (2.9–64.6) | –0.3 (–3.5 to –3.5) |
| Referrals, inc. self-referrals | 29.5 (17.0–42.9) | 13.8 (8.0–21.0) | –15.6 (–25.8 to –4.9) | 33.0 (21.5–45.5) | 14.5 (8.6–21.7) | –18.4 (–27.6 to –9.6) |
| Deaths | 4.9 (0.9–10.0) | 3.9 (0.7–8.3) | –1.0 (–2.8 to –0.1) | 5.4 (1.0–10.9) | 4.7 (0.9–9.3) | –0.8 (–2.2 to 0.0) |
| Life-years lost | 135.7 (25.1–276.9) | 108.7 (19.7–228.5) | –27.0 (–75.9 to –1.6) | 154.8 (29.3–310.1) | 133.1 (24.7–264.6) | –21.7 (–61.7 to –0.2) |
| Cost (2019 US$) | 15729.4 (6368.3–31027.5) | 19297.7 (8413.8–35444.7) | 3568.3 (–8472.2 to 16311.6) | 12508.1 (7056.4–20279.0) | 14525.7 (8603.6–22403.0) | 2017.6 (–5421.3 to 9470.6) |

Quoted as mean (95% quantiles).
*ATT represent the number of children diagnosed with TB who initiate treatment out of 100 children with presumptive TB.
†Indicates different denominators.
ATT, anti-TB treatment; INT, intervention; PHC, primary health care; SOC, standard of care; TB, tuberculosis.

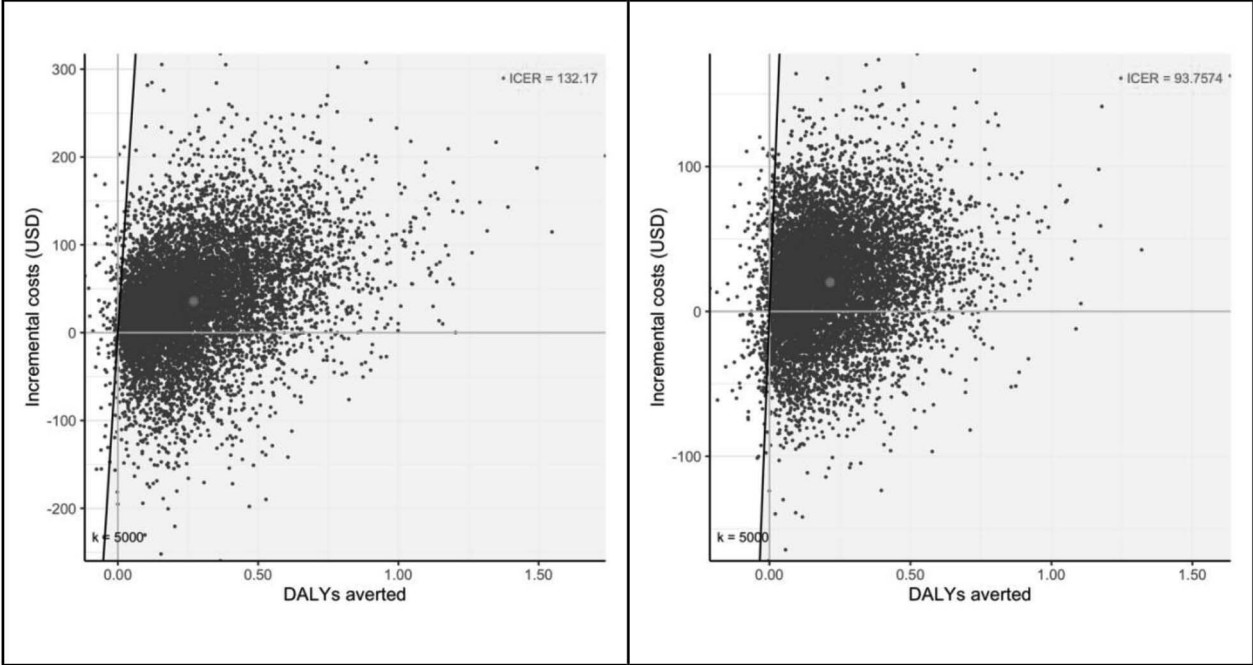

**Figure 2** Cost-effectiveness plane showing the differences in costs (y-axis) and disability-adjusted life-years (DALYs, x-axis) of using the SOS stool method for diagnosis of paediatric TB in Ethiopia (left) and Indonesia (right), compared with standard of care from 10 000 simulations. The grey dot represents the mean incremental costs and DALYs. ICER, incremental cost-effectiveness ratio; k, cost-effectiveness threshold, SOS, simple one-step.

aged <5 years, we found a 14%–20% relative reduction in mortality driven by an increase in sensitivity to detect true TB. However, it is crucial that clinical assessment is still undertaken alongside negative bacteriological test results because bacteriological testing has a low negative predictive value especially in young children.[3 4] Relying on bacteriological testing alone can reduce sensitivity to diagnose true TB and increase mortality, especially if referrals and re-assessments are common under the standard of care (data not shown). We estimated ICERs of US$132 and US$94 per DALY averted in the base-case analyses for Ethiopia and Indonesia, respectively. These ICERs are less than 0.5×GDP, which has been suggested as a rule of thumb for cost-effectiveness thresholds,[28] as well as country-specific estimates of supply-side thresholds.[29 30] The intervention would be especially cost-effective for children under 5 years of age.

Children age <5 years are at higher risk of dying from untreated TB than older children and have the greatest difficulty in spontaneously expectorating sputum. Under the intervention, we found greater increases in bacteriologically diagnosis and greater decreases in referrals in the <5 yearS age group (see age stratified results in online supplemental appendix 3). We found that the cost of introducing the intervention was partially offset by reduced referrals from PHC facilities to hospitals. In Ethiopia, this produced a projected cost saving in the under 5 age group, despite a slight increase in the average total number of assessments done. In taking a healthcare provider's perspective, we did not include patient costs in our analysis, but health-seeking costs are a major driver of catastrophic costs in TB.[31]

There are large uncertainties associated with many parameters describing processes and pathways for paediatric TB. We found no directly applicable estimates of rates of reassessments or (self-)referral at different stages of care, and had to rely on expert opinion. Additionally, the sensitivity and specificity of clinical assessment for paediatric TB is poorly quantified in the literature. Because of this, we placed a particular emphasis on including uncertainty in results, as well as systematically exploring their sensitivity to one-way variation in parameters, and discrete alternative assumptions. For example, because our estimate of true TB prevalence among children with presumptive TB was based on data mainly from hospitals which may have higher prevalence than PHC level, we halved prevalence resulting in increased ICERs by less than a factor of two without changing our qualitative conclusions. Despite these uncertainties, the intervention showed probabilities of being cost-effective >85% in each country across a wide range of cost-effectiveness thresholds. This conclusion was also robust to assuming the SOC used Xpert rather than sputum-smear microscopy at PHC level, and to different choices of discount rate.

Some aspects were deliberately simplified or omitted in the modelling. First, we did not model HIV because paediatric HIV rates in Ethiopia and Indonesia are relatively low at 0.09% and 0.03%,[32] respectively. This may underestimate the benefit from the intervention due to underestimated TB mortality, especially if stool-based

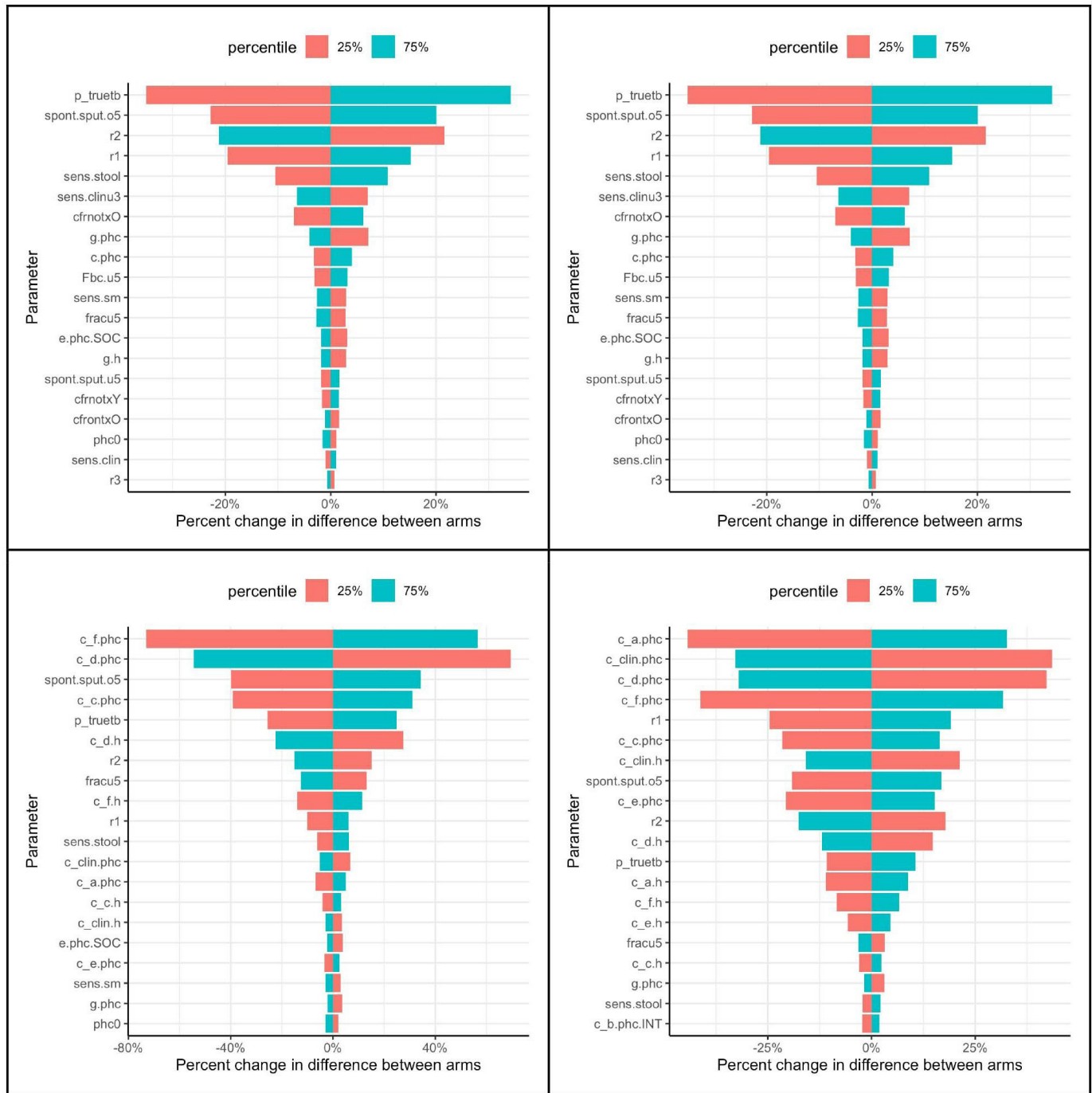

**Figure 3** Tornado plots showing one-way sensitivity of incremental deaths (top row) and incremental costs (bottom row) to parameters for Ethiopia (left) and Indonesia (right). spont.sputo5: spontaneous sputum possible (5–14 years), p_truetb: prevalence of true TB in presumptive, r1: referral from PHC to Hospital after clinical reassessment following bacteriological negative result, r2: referral from PHC to hospital after initial clinical assessment without bacteriological test result, fracu5: fraction of presumptive TB under 5, c_f.phc: cost of TB treatment at PHC after clinical reassessment, c_d.phc: cost of TB treatment at PHC after initial clinical assessment, c_a.phc: cost of clinical and bacteriological TB assessment at PHC, c_clin.h: cost of clinical TB assessment at hospital, c_clin.phc: cost of clinical TB assessment at PHC (only top three parameters on each plot defined here. Please refer to online supplemental appendix 2A,B, for the rest of the parameter definitions. PHC, primary healthcare; SOC, standard of care; TB, tuberculosis.

methods are more effective at diagnosing TB in children with HIV compared with sputum. Second, we did not model drug-resistant TB because of low rates of multidrug-resistant (MDR) TB among new TB cases (all ages) in Ethiopia (1.02% (0.49%–1.54%)) and Indonesia (2.4%

(1.8%–3.3%)). This may underestimate the intervention costs since the higher fractions of cases bacteriologically confirmed via Xpert MTB/Rif mean that more MDR TB will be diagnosed and require more costly second-line treatment. Third, we did not consider the private sector,

which in Indonesia is substantial, and less likely to closely follow guidelines. Our intervention is conceived as being implemented in the public sector, but patients seeking care across both sectors may mean we overestimate the savings to the (public) health system from reduced referrals. Fourth, for pragmatic reasons, country-specific primary cost analyses were not performed and additional one-off programmatic costs for widely introducing Xpert stool testing, such as costs for training and supervision, were not included in our analyses. Both countries are moving to fully replacing SSM by Xpert testing as the primary diagnostic for TB in all patient groups. This may increase logistical issues in both countries which need to be dealt with, such as cartridge shelf life (which is shorter for the Ultra than the G4 cartridge) and module maintenance. Lastly, we modelled the impact of making a stool Xpert-based diagnosis available at the PHC level. The analysis also assumes that all PHC facilities have access to a GeneXpert machine, either on-site or through an effective sample transportation system. Thus, until full access to Xpert testing is available, the coverage of the intervention will be limited.

Furthermore, due to the lack of data from randomised controlled trials and operational studies, we were reliant on early experience to determine acceptability and feasibility of stool-based sampling and diagnostics. Hence, difficulties in implementation that dilute effects or increase costs may be overlooked. However, the recent recommendation to use stool as a primary sample for diagnosing childhood TB[5] has generated interest in Xpert stool testing at national TB programmes. Although we used an illustrative high acceptability rate for stool, this is supported by early experience from the two countries and recently published evidence.[11 33] Apart from the SOS stool method, two other centrifuge-free stool processing methods are being developed,[9 10] which are included in a head-to-head comparison study to compare their performance in diagnosing childhood TB against sputum or gastric aspirate culture. This project has a health economic component, estimating cost-effectiveness of the best performing method. Results of this project are expected at the end of 2021. The TB Speed decentralisation operational research study will report results from use of Xpert on nasopharyngeal aspirate and stool samples at PHC level in early 2022. A small study comparing the SOS stool method to the stool processing kit involved in the head-to-head comparison study[10] concluded that taking into account the sample processing time, consumable requirements and error rates, the SOS stool method would be the method that would be best scalable in low-income and middle-income countries.[34]

Additional evidence from studies and implementation is needed to inform the optimal use of new sample and diagnostic approaches for paediatric TB within real health systems. Studies to quantify referral patterns, the pathways and outcomes of individual patients, and the costs of real-world implementation would be particularly valuable. Further analyses could include context-specific operational research to help design referral systems that best use Xpert machines and minimise cartridge expiry, as well as budget impact analyses to help national programmes plan roll-out and seek funding. Clinical diagnosis remains an important tool for children with TB; helping clinicians diagnose TB in children without bacteriological results or with negative results should be part of intervention design and the role of clinical diagnosis in current and novel diagnostic pathways a topic for further research. The importance of clinical TB diagnosis for children limits the potential impact of bacteriological diagnostics.

## Conclusion

In this modelling analysis, we projected that introduction of routine stool-based Xpert diagnostics at primary healthcare and hospital level would increase the proportion of bacteriologically confirmed TB in children, while reducing child mortality and life-years lost in both Ethiopia and Indonesia. Our analysis suggests that this intervention would be cost-effective in both countries.

**Author affiliations**
[1]ScHARR, The University of Sheffield, Sheffield, UK
[2]Independent consultant, Connect TB, Den Haag, The Netherlands
[3]Global Health and Amsterdam Institute for Global Health and Development, Amsterdam University Medical Center, Amsterdam, The Netherlands
[4]Technical Division, KNCV Tuberculosis Foundation, Den Haag, The Netherlands
[5]Pediatric Department, RSCM Hospital, University of Indonesia Faculty of Medicine, Jakarta, Indonesia
[6]Technical Division, KNCV Tuberculosis Foundation, Addis Ababa, Ethiopia
[7]Department of Paediatrics, Universitas Gadjah Mada Fakultas Kedokteran, Yogyakarta, Indonesia

**Acknowledgements** We thank Anthea Sutton, an Information Specialist at the University of Sheffield, for setting up and conducting the literature database searches; Sarah van de Berg for her assistance in assessing and extracting data from the systematic reviews; Key informants for the expert-solicited parameters from Ethiopia and Indonesia; the NTP and related stakeholders, including pediatricians, scientists, technical agencies and funders implementing childhood TB projects in both countries.

**Contributors** NM, EK, IS, JL, DS, PdH, PJD and EWT conceived the analysis. EK, IS, JL and EWT performed the systematic review. NM, DS and PJD designed and performed the modelling and quantitative analyses. NM, EK, IS, JL, DS, PJD and EWT drafted the manuscript. NK, AB, RT and MG reviewed country epidemiological and health system approaches. NM, PJD and DS had full access to all the data in the study and had final responsibility for the decision to submit for publication. All authors reviewed the overall approach, edited the article and approved the final manuscript.

**Funding** This work was funded by the TB Modelling and Analysis Consortium (Bill & Melinda Gates Foundation, OPP1084276; award to CMY). PJD was supported by a fellowship from the UK Medical Research Council (MR/P022081/1); this UK-funded award is part of the EDCTP2 programme supported by the European Union.

**Competing interests** None declared.

**Patient and public involvement** Patients and/or the public were not involved in the design, or conduct, or reporting, or dissemination plans of this research.

**Patient consent for publication** Not applicable.

**Ethics approval** Not applicable. This was a modelling study based on secondary data.

**Provenance and peer review** Not commissioned; externally peer reviewed.

**Data availability statement** Data are available on reasonable request. Data and code to run these are analyses are available on reasonable request.

**ORCID iDs**
Nyashadzaishe Mafirakureva http://orcid.org/0000-0001-9775-6581
Edine W Tiemersma http://orcid.org/0000-0002-2071-3038

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
