## [Reviewer comments · BMJ Open]

ARTICLE DETAILS

TITLE (PROVISIONAL)	Xpert Ultra stool testing to diagnose tuberculosis in children in Ethiopia and Indonesia: a model-based cost-effectiveness analysis.
AUTHORS	Mafirakureva, Nyashadzaishe; Klinkenberg, Eveline; Spruijt, Ineke; Levy, Jens; Shaweno, Debebe; de Haas, Petra; Kaswandani, Nastiti; Bedru, Ahmed; Triasih, Rina; Gebremichel, Melaku; Dodd, PJ; Tiemersma, Everdina

VERSION 1 – REVIEW

REVIEWER	Cilloni, Lucia Johns Hopkins University, Bloomberg School of Public Health
REVIEW RETURNED	08-Dec-2021

GENERAL COMMENTS	The authors assess the epidemiological impact and cost of utilizing Xpert on stool to detect TB in children, using data from Ethiopia and Indonesia as well as a systematic literature review to inform model parameters. The intervention increased the sensitivity of detecting a true TB case by 10 percentage points and increased the relative number of children initiated on ATT by 19-25%. Among children under 5, the numbers who were bacteriologically investigated increased 30-fold under the intervention scenario. Although costs were higher under the intervention scenario, the intervention was deemed to be cost-effective in both Ethiopia and Indonesia, when using a 0.5xGDP threshold. The research is very interesting and presented mostly in a clear manner. I only have a few comments, that I think could add to the final discussion if addressed and I've pointed out a few inconsistencies as well. Thanks Main comments: - What proportion of individuals across lower-level hospitals and PHC would normally be clinically assessed and diagnosed for TB in the event of a negative microscopy test or in the event of inability to produce sputum? Is your assumption that those who test bac-negative would be followed by a clinical diagnosis close to what would be seen in practice? And what assumption is made for SOC? I couldn't find this information in the paper. The reason I ask this is because I wonder whether, in practice, there may a higher number of clinical diagnoses which may compromise the impact of the intervention (seeing as sensitivity is considerably lower in those clinically diagnosed).- What is the assumed acceptance rate to get tested among those who screen positive and are labelled as presumptive TB cases? Are both SOC and intervention scenarios assuming all those who
---

	can get diagnosed will get diagnosed (i.e. 100% acceptance rate)? If so, how would changes to this parameter affect the analyses? You mention that there's a lack of data on acceptability of stool testing, but would data from other diagnostic tools be helpful in assessing this? For example, uptake Xpert in place of sputum smear microscopy has been quite slow across a number of high burden settings, and I'm wondering whether there may be a threshold of acceptance that makes the intervention not cost-effective? I appreciate these considerations may be outside of the scope of this particular analysis, but they are an important thought when thinking about linking this to implementation.  - Figure captions should be expanded so that they provide a full understanding of what is being shown without the reader having to go back to the text. Figure 2 especially should have a more expansive caption in order to guide the reader through it. Minor corrections  - In caption of Figure 3 (main), there's a typo in the last sentence in bracket, "tope". - On figure 3 (main), there is an incorrect parameter description, "spont.sput.o5" is for children 5/14, not 0-4. - On figure 3 (main), prevalence of TB in presumptive cases which is a top parameter for Indonesia but isn't included in the caption list - on page 7 of appendix 2b, there's an incomplete sentence in the first paragraph "The estimated unit costs..." I think it's missing something along the lines of "for IDN"
--	---

REVIEWER	Mpagama, Stellah Kibong'oto Infectious Diseases Hospital, Medical
REVIEW RETURNED	24-Jan-2022

GENERAL COMMENTS	Mafirakureva et al described a model-based cost effectiveness analysis on the use of XpertMTB/RIF on stool to diagnose tuberculosis in children. The manuscript is well written and authors have done a comprehensive analysis. There is a clear message for the programme and decision makers to decide on the benefit of Xpert in children particularly in non-invasive specimens in this case stool specimen. I see the manuscript meet the criteria for publications and I don't have any comment that needs to be addressed
--

REVIEWER	Siregar, Adiatma Universitas Padjadjaran, Center for Economics and Development Studies, Department of Economics
REVIEW RETURNED	31-Jan-2022

GENERAL COMMENTS	Background  1. Maybe the authors can provide the extent of the issue of the lack of resources to process stool for Xpert testing in developing countries 2. If possible, the authors should also provide several statistics regarding TB and the abovementioned issue in Indonesia and Ethiopia to support the choice of the two countries. This is to assist readers who are not familiar with the condition of the two countries.
--

	Method  1. In regard to costing method, I may have missed this, but I was wondering why authors did not conduct any specific costing study for respective countries. Although the references used by authors are sound enough, a detailed costing study may reveal a more accurate and context specific cost items. This may stem from program costs, staff costs, as well as capital costs such as building etc. I realize this is something that cannot be addressed at this point, but maybe the authors can put some sentences explaining why this is not done in the limitation section. 2. Similarly, authors stated that they use healthcare provider perspective (page 6, line 36). Is this the perspective that is used in the referenced studies? Regardless, healthcare provider may have loose a lot of cost items compared to system perspective, and not to mention the societal perspective. As such, do the costs presented have at least already capture the items borne by other parties aside of the providers? Moreover, authors should state in their limitation section about excluding the costs borne by patients and the subsequent non-healthcare costs as this may have significant portion in access cost from patients' perspective. 3. Authors stated that an ethical clearance is not required as only secondary data is used. However, it is also stated that the study used data from an ongoing study in Ethiopia and Indonesia. I suppose there should be some kind of ethical clearance for this, as they are using sensitive data? 4. Subsequently, it would be good if authors can provide a bit of explanation about the ongoing study in Indonesia and Ethiopia, as it is the main dataset for effectiveness modelling, no? Discussion  1. The authors mentioned that clinical assessment should still be undertaken alongside bacteriological testing. Of this is the case, please mention that this results in additional cost that should be calculated as an integrated part of the test. This can also be an area for future studies. 2. In paragraph 2 of the discussion page 9 line 31, the authors mentioned that they used health system perspective, while in page 6 line 36 they stated that they used provide perspective. Please revise. 3. I suppose the large uncertainties is due to a lot of assumptions used in determining parameters and may have missed the parameters related to local specific context. The authors have provided a well thought sensitivity analysis to overcome this issue, but it is still should be mentioned in the limitation section.
--	--

REVIEWER	Thomas, Tania University of Virginia, Medicine
REVIEW RETURNED	14-Feb-2022

GENERAL COMMENTS	This is an interesting analysis that touches on many important topics related to tackling the struggles with evaluation and management of children with TB. The paper is very well written and careful consideration has been given to the numerous moving parts of the clinical evaluation of TB in children, including 2 separate high-TB-burden settings. The use of two settings increases the applicability of these results. Page 3/105, Line 4: change “recommend” to “recommends” and consider revising “stool on” to “stool testing using”
---

Page 3, Line 23: consider changing “less than half of children...” to “fewer children seeking care at the PHC level needed referral (or self-referred) to higher levels of care” (or “significantly fewer children...”)

Page 5, line 7: Consider removing the parentheses around “young”

Page 5, line 13: Consider rephrasing to “stool testing using Xpert...” or “Xpert testing of stool specimens...”

Page 5, lines 20-22: consider adding performance data on SOS method. This would be very helpful in putting the entire work in more detailed context.

Page 6, line 29: “We assumed that children with a negative bacteriological test under the intervention would receive clinical assessments for TB”—This same assumption should hold for the SOC model, where sputum smear microscopy is not relied on to be an accurate test for TB diagnosis in children and clinical assessments for TB are warranted. To not include the option of clinical assessments for TB in the SOC model carries the risk of falsely favoring the intervention model.

Page 6, lines 38-40. The authors state their hypothesis that the intervention is expected to reduce mortality through higher sensitivity for detecting TB and to reduce referrals and re-assessments. It would be helpful to understand the performance data of the SOS method in detecting MTB from stool specimens, especially in comparison to the data they seem to be using from systematic reviews.

Page 7, lines 5-9: The methods would be strengthened by adding whether the probabilities were obtained from country-specific data. If country-specific data were not used, this should be justified.

Page 8 lines 4-6: Making the initial TB assessment at the PHC level can be challenging, and this is likely to vary based on the age of the child. Adults sometimes require >2 visits and a younger child especially would be no easier to diagnose and have a greater likelihood of requiring 2+ visits and/or referral to a higher levels of care. It is not clear if >2 visits has been incorporated into the sensitivity analyses, but this would be recommended. To give an example, when considering the Figure 3 presented in Appendix 2B, this algorithm may not be applied for all children with suggestive symptoms during the 1st visit and it may require an additional visit (i.e. the 2nd visit) to even consider the use of the algorithm; depending on the findings, “observation for 2 weeks” may be required, necessitating a third visit.

Page 8, lines 34-35: considering Xpert as the universally available bacteriological test in the SOC model carries immense implications for costs related to initial set-up costs/machinery, and training. Have these been accounted for in this scenario?

Pages 6-8/ main methods section: Please include the perspective in the main methods section. Is this CEA from the perspective of the health system? The National TB programs? Or a combination of patient costs + health system costs? I see that it is noted in the CHEERS checklist that this has been included, but I don’t see it in the main manuscript on the page/lines listed.

Pages 6-8/ main methods section: Please include the time horizon in the main methods section. I see that it is noted in the CHEERS checklist that this has been included, but I don’t see it in the main manuscript on the page/lines listed.

Page 9, lines 4-12: please include how the diagnostic test accuracy of the SOS was considered. How was the fraction of children who are able to spontaneously defecate considered? What proportion of children would need to have a separate visit to drop off the stool

specimen? And what costs for this (repeat visit, potentially lost supplies, etc) have been considered?

Page 10/105, lines 15-19: This important point is critical to include and the authors should be applauded for giving it the utmost attention that it deserves.

Page 10, lines 28-29: This sounds like it conflicts with the data presented on page 6 of Appendix 2b regarding the costs of introducing the intervention; please consider revising this statement. Appendix 2b states that costs of training, staff, space, transport and other overhead costs have not been factored in. These are really important costs. As can be seen in the example of running a sputum sample on the GeneXpert platform, the comprehensive costs far exceeds the cost of basic consumables/cartridges.

Page 10, line 31: It would be helpful to introduce the perspective earlier in this paper.

Page 10, line 32: "health-seeking costs are a major driver of catastrophic costs". This statement is likely affected by publication bias and reporting bias. As these authors are aware, the health-system costs are more readily available and measurable. But have the costs related to isolation been measured well for people with microbiologically confirmed TB in these settings? What about parents of young children with TB who have to miss work or otherwise have their daily tasks shifted in order to care for an ill child? The citation listed does consider these indirect/patient-related costs. Consider restructuring this statement

Page 11, lines 22-23: "This assumes..." The work also assumes that all PHCs are able to handle and process stool specimens without additional measures related to training including infection control measures.

Discussion: The discussion spends a considerable amount of space on the uncertainties and limitations, which are helpful. However, the authors may consider additional discussion about the impact on mortality and the thresholds for CE that have been used. All global advocacy groups have been working toward "zero deaths from childhood TB" and the threshold that is used in this work (0.5xGDP) may be considered conservative; the WHO has endorsed using 1xGDP as a threshold.

Table 1, Page 18, lines 24-26: Are the authors including an assumption that 95.3% of people who have bacteriologically positive TB will be lost to follow up prior to initiating treatment? This seems like an alarmingly high number of people who are lost to follow up and requires additional justification. Or is this intended to be a calculation of "1 minus the proportion lost to follow up"? If the latter, please make that more clear because none of the other parameters include an equation. Why are the proportions of loss to follow up equal at the primary health center and referral hospital? Would the assumed extra distance to the referral hospital affect the estimated loss to follow up?

Table 2: The cost of GeneXpert testing seems high in both settings. This publication: <https://pubmed.ncbi.nlm.nih.gov/34695153/> would also support this statement for Ethiopia.

Table 3: The row titled "anti-TB treatments (ATT)" could use clarification, as it is not well understood why the quantity listed in this row would be so much lower than the quantity listed in the following row ("ATT initiated at PHC*"). Clarification about how the

	denominator differs for the latter category may also be helpful for the reader. Table 3: Why is the percent of ATT among true-positives so low in each country? This seems much different than the pre-treatment loss to follow up provided in the parameter estimates. Figure 1: it isn't clear why some boxes are shaded orange while others with the same name/label are not. Figure 3: The abbreviated labels on the Y-axis make this challenging to read.
--	--

VERSION 1 – AUTHOR RESPONSE

Reviewer: 1

Dr. Lucia Cilloni, Johns Hopkins University

Comments to the Author:

The authors assess the epidemiological impact and cost of utilizing Xpert on stool to detect TB in children, using data from Ethiopia and Indonesia as well as a systematic literature review to inform model parameters. The intervention increased the sensitivity of detecting a true TB case by 10 percentage points and increased the relative number of children initiated on ATT by 19-25%. Among children under 5, the numbers who were bacteriologically investigated increased 30-fold under the intervention scenario. Although costs were higher under the intervention scenario, the intervention was deemed to be cost-effective in both Ethiopia and Indonesia, when using a 0.5xGDP threshold.

The research is very interesting and presented mostly in a clear manner. I only have a few comments, that I think could add to the final discussion if addressed and I've pointed out a few inconsistencies as well. Thanks

Thank you for your thoughtful observations and for taking the time to review the manuscript.

Main comments:

- What proportion of clinically diagnosed for TB in the event of a negative microscopy test or in the event of inability to produce sputum? Is your assumption that those who test bac-negative would be followed by a clinical diagnosis close to what would be seen in practice? And what assumption is made for SOC? I couldn't find this information in the paper. The reason I ask this is because I wonder whether, in practice, there may a higher number of clinical diagnoses which may compromise the impact of the intervention (seeing as sensitivity is considerably lower in those clinically diagnosed).

Thank you for these important points and questions. The proportion of individuals diagnosed with TB is a function of the proportion clinically assessed (or reassessed) and the accuracy of clinical assessment. The proportion of children clinically assessed without bacteriological testing was determined by the ability of children in each age category to spontaneously expectorate sputum (i.e 1 minus the proportion able to produce sputum). Data to inform the proportion of clinical reassessment after initial bacteriological exclusion of TB and after short broad-course of antibiotics (both assumed to be 4.5% [IQR; 1.9 - 8.8%]) were based on expert opinion due to unavailability of direct evidence to inform these parameters. We assumed that children with a bacteriological negative test result would always be followed up with clinical assessment under the intervention only. Parameters for the accuracy of TB diagnosis via clinical assessments were assumed to be the same at PHC or hospital and were informed by data on the sensitivity and specificity of clinical diagnosis obtained from our systematic review (although there is little evidence on this). The sensitivity of clinical diagnosis for TB, including among those who test bacteriologically negative, always limits the expected impact from interventions to improve bacteriologic diagnosis - a feature that we have included in our modelling.

Further details on these parameters are available in Appendix 2a under the subsections on 'Accuracy of clinical assessment in bacteriologically negative TB' on Page 13 and 'Summary of other parameters' on Pages 16-18.

- What is the assumed acceptance rate to get tested among those who screen positive and are labelled as presumptive TB cases? Are both SOC and intervention scenarios assuming all those who can get diagnosed will get diagnosed (i.e. 100% acceptance rate)? If so, how would changes to this parameter affect the analyses?

Thank you for this point, which is a very important consideration in planning for and implementing public health interventions. Acceptance rates were factored into the parameter for sample submission, this was assumed as 2.4% in 0-4 years and 38.9% in 5-14 years for sputum as most young children can not spontaneously expectorate, and access to gastric aspirate/induced sputum, etc is limited in these settings. We used actual project data in combination with published studies to estimate these. For stool we assumed up to 91.3% (86.4 - 94.4%) of children will be able to provide a sample, taking into account that some children would not be able to produce stool on-the-spot and parents were unwilling/not able to return to hand in their stool sample; this was also based on our experience in those countries.

You mention that there's a lack of data on acceptability of stool testing, but would data from other diagnostic tools be helpful in assessing this? For example, uptake Xpert in place of sputum smear microscopy has been quite slow across a number of high burden settings, and I'm wondering whether there may be a threshold of acceptance that makes the intervention not cost-effective?

I appreciate these considerations may be outside of the scope of this particular analysis, but they are an important thought when thinking about linking this to implementation.

Thank you for these important points and questions. We agree that the uptake of a new test may be initially slow but incorporation into the WHO recommendations and NTP guidelines would improve uptake. We reviewed stool submission rates for helminths and other tests but could not find routine data on this. We do not expect acceptability of other diagnostic tools to stand in for acceptability of stool testing. We expect the acceptability of stool testing to be higher than for other diagnostic tools used for the diagnosis of TB in children because the alternative samples are obtained using methods that are complex, invasive, stressful, and painful for children. On the other hand, stool is a common sample submitted for investigating sick children (similar to urine) and we do not anticipate any specific issues with its uptake in practice.

Some recent studies (published after our analysis) suggest a high feasibility and acceptability of stool sample collection. de Haas et al. reported high overall submission rates in a proof-of-principle study performed in Ethiopia where 83.7% (123/147) of children were able to provide a stool specimen. Preliminary experiences from the TB-Speed Decentralization and Pneumonia studies, published in the WHO operational handbook on Tuberculosis (24 March 2022), reported that 79.6% (1390/1746) of children enrolled in the decentralization study and 80.7% (944/1170) of children with severe pneumonia were able to provide a stool sample. According to the recently published WHO operational handbook on Tuberculosis, stool has high caregiver acceptability. In our opinion, now that stool testing is recommended by WHO for children with presumptive TB, the acceptance rate would be high in facilities where there is an Xpert machine once it is part of national guidelines. We acknowledge that several other factors are important in the roll-out of a new test, such as slow processes at the NTP, lack of advocacy for updates in guidelines, availability of budget, stocking issues, and issues with machine maintenance. However, these are beyond the scope of the current analysis and were not investigated.

Considering this new evidence, we have added the following sentence, including the new references, in the Discussion on Page 10 Lines 38-40, to support the high acceptance rate assumed in our analysis.

'Although we used an illustrative high acceptability rate for stool, this is supported by our experience in the two countries and recently published studies (de Haas et al. and WHO operational handbook on Tuberculosis).

- Figure captions should be expanded so that they provide a full understanding of what is being shown without the reader having to go back to the text. Figure 2 especially should have a more expansive caption in order to guide the reader through it.

Thank you for these suggestions. We have expanded and improved on the Figure captions as follows:

Figure 1. Simplified diagram of decision-analytic model showing the pathways of care for TB diagnosis and treatment. The decision tree shows children with presumptive TB presenting at either PHC facilities or hospitals where they undergo clinical evaluation with or without bacteriological testing. All children diagnosed with TB are considered for anti-tuberculosis treatment. Children with a negative bacteriological test or those not initially diagnosed with TB after clinical assessment only can be reassessed clinically. Coloured boxes depict the potential of referral to a higher level facility and referrals (indicated by grey lines) from PHC to hospital for further assessment can occur for children without a diagnosis of TB. Each pathway extends to death or survival, however, these details are omitted here to keep the diagram simple. See Appendix 2a for more details on the pathway and parametrization of the model.

MTB: Mycobacterium tuberculosis; TB: tuberculosis; TB Tx: TB diagnosis and anti-TB treatment; PHC: primary health care.

Figure 2. Cost-effectiveness plane showing the differences in costs (y-axis) and disability-adjusted life-years (DALYs, x-axis) of using the SOS stool method for diagnosis of paediatric TB in Ethiopia (left) and Indonesia (right), compared to the standard of care from 10 000 simulations. The red dot represents the mean incremental costs and DALYs.

ICER: incremental cost-effectiveness ratio; k: cost-effectiveness threshold, SoC: Standard of Care.

Minor corrections

- In caption of Figure 3 (main), there's a typo in the last sentence in bracket, "tope".
- On figure 3 (main), there is an incorrect parameter description, "spont.sput.o5" is for children 5/14, not 0-4.
- On figure 3 (main), prevalence of TB in presumptive cases which is a top parameter for Indonesia but isn't included in the caption list

Thank you for picking up these issues. We have made the necessary changes in the figure captions as provided below.

Figure 3. Tornado plots showing one-way sensitivity of incremental deaths (top row) and incremental costs (bottom row) to parameters for Ethiopia (left) and Indonesia (right). `spont.sputo5`: spontaneous sputum possible (5-14 years), `p_truetb`: prevalence of true TB in presumptive, `r1`: referral from PHC to Hospital after clinical re-assessment following bacteriological negative result, `r2`: referral from PHC to Hospital after initial clinical assessment without bacteriological test result, `fracu5`: fraction of presumptive TB under 5, `c_f.phc`: cost of TB treatment at PHC after clinical re-assessment, `c_d.phc`: cost of TB treatment at PHC after initial clinical assessment, `c_a.phc`: cost of clinical and bacteriological TB assessment at PHC, `c_clin.h`: cost of clinical TB assessment at Hospital, `c_clin.phc`: cost of clinical TB assessment at PHC (Only top 3 parameters on each plot defined here).

Please refer to Appendix 2a, Tables A7-12 and Appendix 2b, Table A1 for the rest of the parameter definitions).

- on page 7 of appendix 2b, there's an incomplete sentence in the first paragraph "The estimated unit costs..." I think it's missing something along the lines of "for IDN"

We have made the necessary changes on Page 7 of Appendix 2b to make the sentence complete as follows:

'The estimated unit costs for the GeneXpert test are \$26.04 (95% UI; 18.95-33.13) for ETH and \$23.70 (95% UI; 16.59-30.81) for IDN'.

Reviewer: 2

Dr. Stellah Mpagama, Kibong'oto Infectious Diseases Hospital

Comments to the Author:

Mafirakureva et al described a model-based cost effectiveness analysis on the use of XpertMTB/RIF on stool to diagnose tuberculosis in children. The manuscript is well written and authors have done a comprehensive analysis. There is a clear message for the programme and decision makers to decide on the benefit of Xpert in children particularly in non-invasive specimens in this case stool specimen. I see the manuscript meet the criteria for publications and I don't have any comment that needs to be addressed

Many thanks for reviewing our manuscript and for these thoughtful observations.

Reviewer: 3

Dr. Adiatma Siregar, Universitas Padjadjaran

Comments to the Author:

Background

1. Maybe the authors can provide the extent of the issue of the lack of resources to process stool for Xpert testing in developing countries
2. If possible, the authors should also provide several statistics regarding TB and the above mentioned issue in Indonesia and Ethiopia to support the choice of the two countries. This is to assist readers who are not familiar with the condition of the two countries.

Many thanks for these important points. We acknowledge that most developing countries have limited resources and the SOS stool method was specifically designed to be simple and cheap. As highlighted in the introduction, the SOS stool method 'can be applied in any laboratory with an Xpert machine, as it does not require additional equipment or consumables than those delivered routinely with the Xpert cartridges'. We also acknowledge that assuming that all facilities have access to a GeneXpert machine, either on-site or through an effective sample transportation system, may not correctly represent the situation in the two countries. We did not model system constraints because this was beyond the scope of the current analysis.

We have added some statistics on the epidemiology of TB and public/private healthcare split in the two countries included in the analysis on Page 4 lines 39-44.

'Ethiopia and Indonesia are currently among the 30 high TB burden countries in the world (World Health Organization, 2021). The incidence of TB was estimated to be 301 (276-328) per 100,000 population with 824 000 (755 000-897 000) people falling ill with TB in 2020 in Indonesia. In Ethiopia, the incidence of TB was estimated to be 132 (92-178) per 100,000 population with 151 000 (106 000-

205 000) people falling ill with TB in 2020. While TB diagnosis and treatment in Ethiopia largely occur in the public sector, the private sector plays a substantial role in Indonesia’.

Method

1. In regard to costing method, I may have missed this, but I was wondering why authors did not conduct any specific costing study for respective countries. Although the references used by authors are sound enough, a detailed costing study may reveal a more accurate and context specific cost items. This may stem from program costs, staff costs, as well as capital costs such as building etc. I realize this is something that cannot be addressed at this point, but maybe the authors can put some sentences explaining why this is not done in the limitation section.

We completely agree that cost analyses performed using primary data from the two countries would have better informed the cost parameters in our model. It was not feasible to perform such studies to inform our analysis due to resource and time limitations hence our reliance on publicly available sources for cost data. However, our analysis leveraged publicly available sources of data for the two countries to estimate country-specific costs, whenever available. As such all the main costs included in the analysis including facility visits for assessments and treatment, sputum smear microscopy examination, and gene Xpert test were largely informed by country-specific data. We have now explicitly stated the lack of primary cost analyses as a potential limitation in the manuscript on Page 8 lines 23-26.

‘Fourthly, for pragmatic reasons, country-specific primary cost analyses were not performed and additional one-off programmatic costs for widely introducing Xpert stool testing, such as costs for training and supervision, were not included in our analyses’.

2. Similarly, authors stated that they use healthcare provider perspective (page 6, line 36). Is this the perspective that is used in the referenced studies? Regardless, healthcare provider may have loose a lot of cost items compared to system perspective, and not to mention the societal perspective. As such, do the costs presented have at least already capture the items borne by other parties aside of the providers? Moreover, authors should state in their limitation section about excluding the costs borne by patients and the subsequent non-healthcare costs as this may have significant portion in access cost from patients’ perspective.

Thank you for your comments. We used a healthcare provider perspective for our analysis because the aim of our analysis was to provide evidence (to policy/decision-makers) on the impact and cost-effectiveness of adding the SOS stool method to the national algorithms for diagnosing TB in children that can serve as an aid to decision-making. We, therefore, included costs incurred from the health service that would be relevant to answering this question. However, we explicitly indicated that we did not include patient costs and highlighted their potential impact on patient-care seeking on Page 9 lines 32-33 of the Discussion where we state that:

‘In taking a healthcare provider’s perspective, we did not include patient costs in our analysis, but acknowledge that health-seeking costs are a major driver of catastrophic costs in TB.’

3. Authors stated that an ethical clearance is not required as only secondary data is used. However, it is also stated that the study used data from an ongoing study in Ethiopia and Indonesia. I suppose there should be some kind of ethical clearance for this, as they are using sensitive data?

Thank you for this point. For these studies, indeed, ethical clearance was obtained. However, we only used summary measures that do not require ethics and did not use any patient-specific data in the analyses we present here, so we deemed it not necessary to include ethical clearance reference numbers.

4. Subsequently, it would be good if authors can provide a bit of explanation about the ongoing study in Indonesia and Ethiopia, as it is the main dataset for effectiveness modelling, no?

Thank you for this point. In fact, these datasets did not form the core dataset for this modelling study, but only provided information on the proportion of children who would be able to provide a spontaneous sputum sample. Most data used in this study come from publicly available sources referred to in the paper or the appendices, or from assumptions validated through expert opinion.

Discussion

1. The authors mentioned that clinical assessment should still be undertaken alongside bacteriological testing. Of this is the case, please mention that this results in additional cost that should be calculated as an integrated part of the test. This can also be an area for future studies.

Thank you for this point. Routine practice for the diagnosis of TB involves clinical assessment with or without bacteriological testing. Thus, clinical assessment is done both under the SOC and in the alternative scenarios included in our study/estimations. Our discussion point aimed to reiterate the importance of clinical assessment even in the presence of bacteriological testing because microbiological confirmation is only possible in a fraction of children with TB and a negative test does not exclude TB, especially in young children. Because clinical testing is almost always performed, this will not result in any additional costs beyond those already included in the model.

2. In paragraph 2 of the discussion page 9 line 31, the authors mentioned that they used health system perspective, while in page 6 line 36 they stated that they used provide perspective. Please revise.

Thank you for picking up this inconsistency. We have made the necessary changes to be consistent with the 'healthcare provider's perspective' throughout the manuscript.

3. I suppose the large uncertainties is due to a lot of assumptions used in determining parameters and may have missed the parameters related to local specific context. The authors have provided a well thought sensitivity analysis to overcome this issue, but it is still should be mentioned in the limitation section.

Thank you for this point and for your thoughtful observations. We recognized potential uncertainties associated with many parameters in the model hence our decision to treat all parameters as uncertain early on in the modelling process. This allowed us to explore the importance and potential impact of these uncertainties on model outcomes. Our results were robust to these parameter uncertainties. We also discussed these parameter uncertainties at reasonable length in the third paragraph of the Discussion section, including the following text that is already included in the manuscript:

'There are large uncertainties associated with many parameters describing processes and pathways for paediatric TB. We found no directly applicable estimates of rates of reassessments or (self)referral at different stages of care, and had to rely on expert opinion. Additionally, the sensitivity and specificity of clinical assessment for paediatric TB is poorly quantified in the literature. Because of this, we placed a particular emphasis on including uncertainty in results, as well as systematically exploring their sensitivity to one-way variation in parameters, and discrete alternative assumptions. For example, because our estimate of true TB prevalence among children with presumptive TB was based on data mainly from hospitals which may have higher prevalence than PHC level, we halved prevalence resulting in increased ICERs by less than a factor of two without changing our qualitative conclusions'.

Reviewer: 4
Dr. Tania Thomas, University of Virginia

Comments to the Author:

This is an interesting analysis that touches on many important topics related to tackling the struggles with evaluation and management of children with TB. The paper is very well written and careful consideration has been given to the numerous moving parts of the clinical evaluation of TB in children, including 2 separate high-TB-burden settings. The use of two settings increases the applicability of these results. There are some additional suggestions and clarifications that may strengthen this work (see attachment).

Thank you for these thoughtful observations and for taking the time to review the manuscript.

Page 3/105, Line 4: change “recommend” to “recommends” and consider revising “stool on” to “stool testing using”

We have made the suggested changes in the manuscript as follows:

‘The World Health Organization currently recommends stool on GeneXpert MTB/Rif (Xpert) for the diagnosis of paediatric tuberculosis (TB)’.

Page 3, Line 23: consider changing “less than half of children...” to “fewer children seeking care at the PHC level needed referral (or self-referred) to higher levels of care” (or “significantly fewer children...”)

We have made the suggested changes in the manuscript as follows:

‘Under the intervention, fewer children seeking care at PHC were referred (or self-referred) to higher levels of care; the number of children initiating anti-TB treatment (ATT) increased by 18-25%, and more children (85%) initiated ATT at PHC level’.

Page 5, line 7: Consider removing the parentheses around “young”

We have removed the parentheses as suggested in the manuscript as follows:

‘Partly, this is because the main specimen used for diagnosing pulmonary TB is sputum, which is challenging to obtain, especially from young children’.

Page 5, line 13: Consider rephrasing to “stool testing using Xpert...” or “Xpert testing of stool Specimens...”

We have made the suggested changes in the manuscript as follows:

‘Since January 2020, WHO recommends Xpert testing of stool specimens as a primary diagnostic test for TB in children with signs and symptoms of pulmonary TB’.

Page 5, lines 20-22: consider adding performance data on SOS method. This would be very helpful in putting the entire work in more detailed context.

Thank you for this point. Currently, evidence on the diagnostic accuracy of the SOS stool method is still being collected and thus limited. However, data from a systematic review by Mesman et al. of Xpert MTB/RIF testing on stool specimens reported pooled sensitivity and specificity of stool Xpert of 57.1% (95% confidence interval (CI) 51.5-62.7%) and 98.1% (CI 97.5-98.6%), respectively, compared

to culture on a respiratory sample as the reference standard. Our analysis assumed the same accuracy for the SOS method. A systematic review by MacLean et al. reported a pooled sensitivity of 67% (95%CI, 52-79%) compared to a combination of sputum culture and sputum Xpert as the reference standards. Another systematic review by Gebre et al. reported a pooled sensitivity of 50% (95%CI, 44-56%) compared to culture on a respiratory sample as the reference standard. A preliminary analysis of the performance of Xpert MTB/RIF Ultra on stool samples among children with presumptive TB from a head-to-head comparison of three stool processing methods (recently published in the WHO operational handbook on Tuberculosis) reported comparable sensitivity and specificity for the SOS method on Xpert Ultra of 52.1% (95% CI; 38.3–65.5%) and 97.5% (95% CI; 94.9–98.9%), respectively.

Details on the performance of Xpert MTB/RIF Ultra on stool samples used for this analysis are provided in the technical Appendix 2a: Model parameter estimation on the 'Accuracy of bacteriological tests' subsection on Pages 14-15. We have also added the following text to the manuscript on Page 4 Lines 22-28 to help clarify this point:

'Limited preliminary data suggest that Ultra on stool samples processed using the SOS stool method has higher sensitivity compared to other stool processing methods. Available systematic reviews on the diagnostic accuracy of stool testing have reported a sensitivity of 50-67% (Gebre et al., MacLean et al. & Mesman et al.). The variation in sensitivity estimates may be explained by a variation in studies included, and thus, variation in study populations, stool processing methods, and reference standards (sputum culture (Gebre et al. & Mesman et al.) or a combination of sputum culture and sputum Xpert (MacLean et al.)) included in each review'.

Page 6, line 29: "We assumed that children with a negative bacteriological test under the intervention would receive clinical assessments for TB"—This same assumption should hold for the SOC model, where sputum smear microscopy is not relied on to be an accurate test for TB diagnosis in children and clinical assessments for TB are warranted. To not include the option of clinical assessments for TB in the SOC model carries the risk of falsely favoring the intervention model.

Thank you for these comments. Our modelling assumed that only a small proportion of children with a negative bacteriological test, 4.5% (IQR; 1.9 - 8.8%), would receive clinical assessments for TB in both the SOC while all children with a negative bacteriological test were assumed to receive clinical assessments in the intervention. We have revised the assumption to include the SOC as follows;

'We assumed that children with a negative bacteriological test under the intervention would receive clinical assessments for TB, while only a small proportion would get clinical assessments under SOC'.

We have also included the following statement in the Discussion to further highlight the importance of clinical diagnosis of TB in children.

'The importance of clinical TB diagnosis for children limits the potential impact of bacteriological diagnostics'.

Page 6, lines 38-40. The authors state their hypothesis that the intervention is expected to reduce mortality through higher sensitivity for detecting TB and to reduce referrals and re-assessments. It would be helpful to understand the performance data of the SOS method in detecting MTB from stool specimens, especially in comparison to the data they seem to be using from systematic reviews.

Thank you for this point. As discussed in an earlier point above, we assumed that the SOS method's performance would be similar to that of data from a systematic review by Mesman et al. of Xpert MTB/RIF testing on stool specimens: reported pooled sensitivity and specificity of stool Xpert of

57.1% (95% confidence interval (CI) 51.5-62.7%) and 98.1% (CI 97.5-98.6%), respectively, compared to culture on a respiratory sample as the reference standard. These data are comparable preliminary estimates of sensitivity and specificity for the SOS method on Xpert Ultra of 52.1% (95% CI; 38.3–65.5%) and 97.5% (95% CI; 94.9–98.9%), respectively, recently published in the WHO operational handbook on Tuberculosis. We have also added the following text to the manuscript to further highlight this assumption on Page 5 lines 38-40.

'The accuracy of Xpert testing on stool using the SOS method was modelled based on a systematic review Mesman et al. which reported pooled sensitivity and specificity of stool Xpert of 57.1% (95% confidence interval (CI) 51.5-62.7%) and 98.1% (CI 97.5-98.6%), respectively, compared to culture on a respiratory sample as the reference standard'.

Page 7, lines 5-9: The methods would be strengthened by adding whether the probabilities were obtained from country-specific data. If country-specific data were not used, this should be justified.

Thank you for this point. Specific details on model parameterization and sources are given under the 'Literature review and parameterization' section, where we have clearly described the process of identifying model parameters and justification for the different sources used. In addition, details of all parameter assumptions as well as their sources and for which countries they apply are outlined in detail in the technical appendix 2a. We have added some new text and revised the current text (under Results line 3) to highlight the probabilities that were informed by country-specific data from our ongoing studies or published studies as suggested by the reviewer:

'Country-specific data was used to inform the proportion of children submitting a spontaneously expectorated sputum sample, the fraction of presumptive TB in children under 5 years, and the level of initial care-seeking at PHC. We used existing systematic reviews for the basis of parameters describing diagnostic test accuracy, our own pooled estimates of true TB prevalence among presumptive TB, the fraction of TB that is bacteriological confirmable, and the fraction of children able to spontaneously expectorate'.

Page 8 lines 4-6: Making the initial TB assessment at the PHC level can be challenging, and this is likely to vary based on the age of the child. Adults sometimes require >2 visits and a younger child especially would be no easier to diagnose and have a greater likelihood of requiring 2+ visits and/or referral to a higher levels of care. It is not clear if >2 visits has been incorporated into the sensitivity analyses, but this would be recommended. To give an example, when considering the Figure 3 presented in Appendix 2B, this algorithm may not be applied for all children with suggestive symptoms during the 1st visit and it may require an additional visit (i.e. the 2nd visit) to even consider the use of the algorithm; depending on the findings, "observation for 2 weeks" may be required, necessitating a third visit.

Thank you for raising this important point which we also discussed at length at the time of the analysis and writing. We consulted in-country TB experts who highlighted the variability in the number of visits needed for a TB diagnosis. The experts also indicated that most children presenting at PHC are often referred directly to the hospital after the first visit (80% in Ethiopia and 50% in Indonesia), unless broad-spectrum antibiotics have been prescribed to see if this leads to complaints resolving. The experts confirmed that the number of visits can range from 1 to a maximum of 3 in most instances. We used 2 visits for our analysis, which we think is a reasonable compromise between the two extremes. This is actually a conservative approach because modelling a higher number of visits would improve cost-effectiveness through cost savings gained by reduced referrals to hospitals.

Page 8, lines 34-35: considering Xpert as the universally available bacteriological test in the SOC model carries immense implications for costs related to initial set-up costs/machinery, and training. Have these been accounted for in this scenario?

Thank you for this point. We completely agree with your comment. We assumed the use of sputum smear microscopy in the base case analysis because of the known limited availability of Xpert machines. We only considered the universal availability of Xpert in the sensitivity analysis. We also acknowledged in the Discussion that this analysis did not account for costs associated with the widespread implementation of Xpert machines and further highlighted this as an area for potential future research.

Pages 6-8/ main methods section: Please include the perspective in the main methods section. Is this CEA from the perspective of the health system? The National TB programs? Or a combination of patient costs + health system costs? I see that it is noted in the CHEERS checklist that this has been included, but I don't see it in the main manuscript on the page/lines listed.

Thank you for this suggestion. We have already stated that we used the 'healthcare provider perspective' for the analysis in the main methods section under the subsection on 'Cost parameters and health economic approach' on page 7 line 2 (as noted in the CHEERS checklist) which states that:

'We collected costs (reported in 2019 USD) from the healthcare provider's perspective...'

Pages 6-8/ main methods section: Please include the time horizon in the main methods section. I see that it is noted in the CHEERS checklist that this has been included, but I don't see it in the main manuscript on the page/lines listed.

Thank you for this comment. We already included the time horizon for the cost analysis in the main methods section of the manuscript under the subsection on 'Cost parameters and health economic approach' on page 7 line 5 (as noted in the CHEERS checklist) which states that:

'Costs were assumed to accrue in the present, with no discounting applied'.

We have now also included the time horizon for the outcomes (life-years and DALYs) in the main methods section of the manuscript under the subsection on 'Cost parameters and health economic approach' on page 7 lines 24-26 (as also noted in the CHEERS checklist) which states that:

'We used a disability-adjusted life-year (DALY) framework, calculating the life-years saved over a lifetime horizon with a discount rate of 3% based on United Nations Population Division country-specific life tables'.

Page 9, lines 4-12: please include how the diagnostic test accuracy of the SOS was considered. How was the fraction of children who are able to spontaneously defecate considered? What proportion of children would need to have a separate visit to drop off the stool specimen? And what costs for this (repeat visit, potentially lost supplies, etc) have been considered?

Thank you for this point. We assumed that up to 95% of children would be able to provide a stool sample, taking into account that some children would not be able to produce stool on-the-spot and parents were unwilling/not able to return to hand in their stool sample; this was also based on our experience in those countries. We acknowledge that some children would require an additional visit to submit the stool sample, but we assumed this will consume very minimal healthcare provider

resources resulting in insignificant costs. Patients may incur costs for the additional visit however patient costs were not modelled in this analysis.

The diagnostic test accuracy of the SOS method assumed for this analysis was based on a systematic review of Xpert MTB/RIF testing on stool specimens which reported pooled sensitivity and specificity of stool Xpert of 57.1% (95% confidence interval (CI) 51.5-62.7%) and 98.1% (CI 97.5-98.6%), respectively, compared to culture on a respiratory sample as the reference standard.

Page 10/105, lines 15-19: This important point is critical to include and the authors should be applauded for giving it the utmost attention that it deserves.

Thank you for your thoughtful observations.

Page 10, lines 28-29: This sounds like it conflicts with the data presented on page 6 of Appendix 2b regarding the costs of introducing the intervention; please consider revising this statement. Appendix 2b states that costs of training, staff, space, transport and other overhead costs have not been factored in. These are really important costs. As can be seen in the example of running a sputum sample on the GeneXpert platform, the comprehensive costs far exceeds the cost of basic consumables/cartridges.

Thank you for this point. We fully agree on the importance of these costs and can confirm that we included staff, consumables/cartridges, equipment, and overheads in the cost of GeneXpert test. These details are provided in Appendix 2b, Overview of cost parameters, under the GeneXpert test section. We also acknowledge the importance of costs associated with staff training, especially when widely introducing public health intervention, which we did not include in our cost analysis. We already indicated this as a potential limitation of the analysis in the manuscript using the following text which we have rephrased to capture other comments above:

'Fourthly, for pragmatic reasons, country-specific primary cost analyses were not performed and additional one-off programmatic costs for widely introducing Xpert stool testing, such as costs for training and supervision, were not included in our analyses'.

Page 10, line 31: It would be helpful to introduce the perspective earlier in this paper.

Thank you for this point. We introduced the perspective earlier in this paper in the main methods section under the subsection on 'Cost parameters and health economic approach' on page 7 line 2 (as noted in the CHEERS checklist) which states that:

'We collected costs (reported in 2019 USD) from the healthcare provider's perspective...'

In addition, we have changed the text 'health system perspective' to 'healthcare provider perspective' to improve consistency throughout the manuscript.

Page 10, line 32: "health-seeking costs are a major driver of catastrophic costs". This statement is affected by publication bias and reporting bias. As these authors are aware, the health-system costs are more readily available and measurable. But have the costs related to isolation been measured well for people with microbiologically confirmed TB in these settings? What about parents of young children with TB who have to miss work or otherwise have their daily tasks shifted in order to care for an ill child?

The citation listed does consider these indirect/patient-related costs. Consider restructuring this statement

Thank you for this point. The statement 'health-seeking costs are a major driver of catastrophic costs' is referring to costs incurred by patients as they seek medical care for their illnesses and does not refer to 'healthcare system costs'. As previously discussed in responses to other comments, the analysis focussed on healthcare provider costs only and did not include any costs incurred by patients or their households (parents in this case). In addition, our analysis focused on drug-susceptible TB and therefore we did not include any costs related to the isolation of patients that may be required for drug-resistant TB. This is highlighted in the Methods under the Modelling approach on Page 6 Line 11 and Discussion on Page 10, lines 16-20.

Page 11, lines 22-23: "This assumes..." The work also assumes that all PHCs are able to handle and process stool specimens without additional measures related to training including infection control measures.

We have made the suggested changes in the manuscript as follows:

'The analysis also assumes that all PHC facilities have access to a GeneXpert machine, either on-site or through an effective sample transportation system'.

Discussion: The discussion spends a considerable amount of space on the uncertainties and limitations, which are helpful. However, the authors may consider additional discussion about the impact on mortality and the thresholds for CE that have been used. All global advocacy groups have been working toward "zero deaths from childhood TB" and the threshold that is used in this work (0.5xGDP) may be considered conservative; the WHO has endorsed using 1xGDP as a threshold.

Thank you for these important points. There is an ongoing methodological debate on the choice of appropriate thresholds for cost-effectiveness analysis and no consensus has been reached on the subject yet. The 1-3 x GDP per capita threshold is often used as a guide in countries where explicit cost-effectiveness thresholds are lacking. In recent years, the WHO has distanced itself from the 1-3 x GDP per capita threshold (Bertram et al, 2016). The recent work of Woods et al., Ochalek et al., and others is a relatively crude first attempt to "reverse engineer" the implied thresholds used in different settings based on their current healthcare provision, i.e. to give a sense of what ICERs would be likely to be "competitive" within those health systems. These different authors suggested that 1-3 x GDP per capita is too high as an estimate of marginal ICERs in current health systems. In the absence of explicit thresholds in the two countries we evaluated and in light of recent developments in the area, we used the 0.5 x GDP per capita, which is reported as a crude summary of econometric work to estimate typical marginal ICERs in health systems. We also present cost-effectiveness acceptability curves (Appendix 3 Figures A1 and A2) to provide decision-makers with probabilities of interventions being cost-effective at their preferred threshold.

The implementation of the SOS method is projected to have a substantial impact on mortality, however, this is likely to depend on a number of factors including uptake rates, coverage, and availability of Xpert machines, all of which were not assessed in the current analysis.

Table 1, Page 18, lines 24-26: Are the authors including an assumption that 95.3% of people who have bacteriologically positive TB will be lost to follow up prior to initiating treatment? This seems like an alarmingly high number of people who are lost to follow up and requires additional justification. Or is this intended to be a calculation of "1 minus the proportion lost to follow up"? If the latter, please make that more clear because none of the other parameters include an equation. Why are the proportions of loss to follow up equal at the primary health center and referral hospital? Would the assumed extra distance to the referral hospital affect the estimated loss to follow up?

Thank you for this point. It is indeed intended to be a calculation of '1-the proportion of children lost to follow up'. We have renamed the parameter as 'proportion of bacteriologically confirmed children initiating anti-TB treatment' to improve clarity and avoid any misunderstanding. The proportions of loss to follow-up at the primary health center and referral hospital were assumed to be equal because data to inform these parameters were not available. Our consultations with country experts did not give rise to any suggestions to change these assumptions.

Table 2: The cost of GeneXpert testing seems high in both settings. This publication: <https://pubmed.ncbi.nlm.nih.gov/34695153/> would also support this statement for Ethiopia.

Thank you for this point and for sharing the Kaso et al. publication which we have carefully reviewed. We completely agree that our estimates are higher than the estimate reported in this publication, and this is largely due to methodological differences. Firstly, our analysis used a unit cost for the Xpert cartridge of \$11.80 available from the OneHealth Costing Tool, which is higher than the \$9.98 used in the Kaso et al. publication. We assumed the OneHealth figure included additional costs such as freight and clearances (which we think are not included in the \$9.98 cost) hence our decision to use it in our analysis. Secondly, we included an annual maintenance cost for the Xpert machine, something which Kaso et al. did not include. Lastly, our analysis incorporated costs associated with unused Xpert equipment capacity by accounting for (through modelling) the actual number of tests performed per day in relation to an assumed daily maximum capacity of 16 tests for the Xpert MTB/RIF 4-module machine. On the contrary, it is not immediately clear whether and how Kaso et al. considered equipment capacity in their analysis. We think all these methodological choices led to a more complete but justifiably higher estimate of the cost of Xpert testing in our analysis.

Table 3: The row titled "anti-TB treatments (ATT)" could use clarification, as it is not well understood why the quantity listed in this row would be so much lower than the quantity listed in the following row ("ATT initiated at PHC**"). Clarification about how the denominator differs for the latter category may also be helpful for the reader.

Thank you for this point. We have now included an explanatory remark in the manuscript to clarify what anti-TB treatments mean as shown below:

'Table 3 Outcomes per 100 children seeking care (*asterisk indicates different denominators) under the standard of care (SOC) and intervention (INT) in each country. Quoted as Mean (95% quantiles). ϕ anti-TB treatments (ATT) represent the number of children diagnosed with TB who initiate treatment out of 100 children with presumptive TB. TB = tuberculosis; ATT = anti-TB treatment; PHC = primary health care; USD = United States Dollars'.

Table 3: Why is the percent of ATT among true-positives so low in each country? This seems much different than the pre-treatment loss to follow up provided in the parameter estimates.

Thank you for this point. The outcome 'percent of true- positive receiving ATT' was intended to refer to 'percent of true TB receiving ATT'. We have made this change in the manuscript to improve clarity.

Figure 1: it isn't clear why some boxes are shaded orange while others with the same name/label are Not.

Thank you for this point. We have since expanded and improved on the captions for Figure 1, including a description of the coloured boxes as follows:

'Coloured boxes depict the potential of referral to a higher level facility'.

Figure 3: The abbreviated labels on the Y-axis make this challenging to read.

Thank you for this point. We have increased the font size for the abbreviated Y-axis labels on the revised Figure 3 to improve readability.

VERSION 2 – REVIEW

REVIEWER	Cilloni, Lucia Johns Hopkins University, Bloomberg School of Public Health
REVIEW RETURNED	09-May-2022

GENERAL COMMENTS	Thank you for addressing my comments, I am pleased with the edits made to the manuscript. Thank you again for this important piece of work.
---

REVIEWER	Siregar, Adiatma Universitas Padjadjaran, Center for Economics and Development Studies, Department of Economics
REVIEW RETURNED	27-Apr-2022

GENERAL COMMENTS	The authors have made the necessary revision and responses in line with my previous comments. I have no further revision.
---